# XPro-Design: Rational Protein Engineering Framework Using Explainable AI

## Abstract

Protein engineering seeks to rationally tailor proteins to achieve specific structural and functional objectives. These objectives encompass enhancing catalytic efficiency, modifying substrate specificity, improving binding affinity, reducing immunogenicity, and increasing stability under adverse conditions. A major bottleneck is protein instability, as elevated temperatures often drive degradation and compromise activity. Developing thermostable proteins is therefore a key objective in engineering efforts. Here, we present XPro-Design, an explainable AI driven framework for protein optimization that integrates amino acid-level explanations of functional impact into generative modeling. Our method captures epistatic interactions and the mutational landscape by training a low-rank matrix, which biases the generative model toward high-scoring regions of sequence space. This enables targeted generation of candidate variants optimized for thermostability, while remaining extensible to other objectives. XPro-Design further uses distribution tempering and annealing to effectively balance exploration vs exploitation without compromising on structural integrity. We demonstrate rational, causality driven design of protein variants with melting temperatures nearly 2x that of their wild-type counterparts, while preserving binding pocket integrity and domain architecture. Moreover, engineered variants show up to 38% lower folding free energy relative to wild-type indicating significantly enhanced thermodynamic stability. XPro-Design establishes a generalizable strategy for explainable and controllable protein design, enabling multi-objective optimization beyond thermostability.

## 1 Introduction

Protein engineering is a cornerstone of modern biotechnology, enabling the rational tailoring of proteins for therapeutic, industrial, and synthetic biology applications. Engineered proteins can improve binding affinity, reduce immunogenicity, extend half-life, function in harsh environments, catalyze reactions with higher efficiency, or perform novel tasks such as biosensing and pathway modulation. Despite these diverse applications, a central challenge remains: amino acid mutations often exert complex, non-additive effects on structure and function, making the sequence–function landscape difficult to navigate. Among targeted properties, thermostability is especially critical, as proteins unstable at elevated temperatures readily unfold, aggregate, and lose function. Stabilizing determinants include hydrophobic core packing, hydrogen-bond networks, covalent linkages such as disulfide bonds, and minimization of unfavorable electrostatic or solvent-exposed hydrophobic interactions, whereas disruptions to these features often destabilize proteins. Balancing these opposing contributions defines the mutational landscape of thermostability and underscores the need for methods that accurately capture and exploit sequence–structure–function relationships.

Protein stability prediction has been approached through both physics-based and machine learning methods. Classical tools such as FoldX (Schymkowitz et al., 2005) and Rosetta (Leaver-Fay et al., 2011; Fleishman et al., 2011; Leman et al., 2020) estimate mutational effects on folding free energy by modeling structural energetics and have long served as reference points for benchmarking. More recent data-driven approaches, including DDGun/DDGun3D (Montanucci et al., 2019), and DDGemb (Savojardo et al., 2025), leverage evolutionary features or embeddings from protein language models (Rives et al., 2019; Lin et al., 2022) to predict $\Delta\Delta G$ of mutation. Other efforts such as DeepTM (Li et al., 2023), ProTstab2 (Yang et al., 2022), and DeepSTABp (Jung et al., 2023)

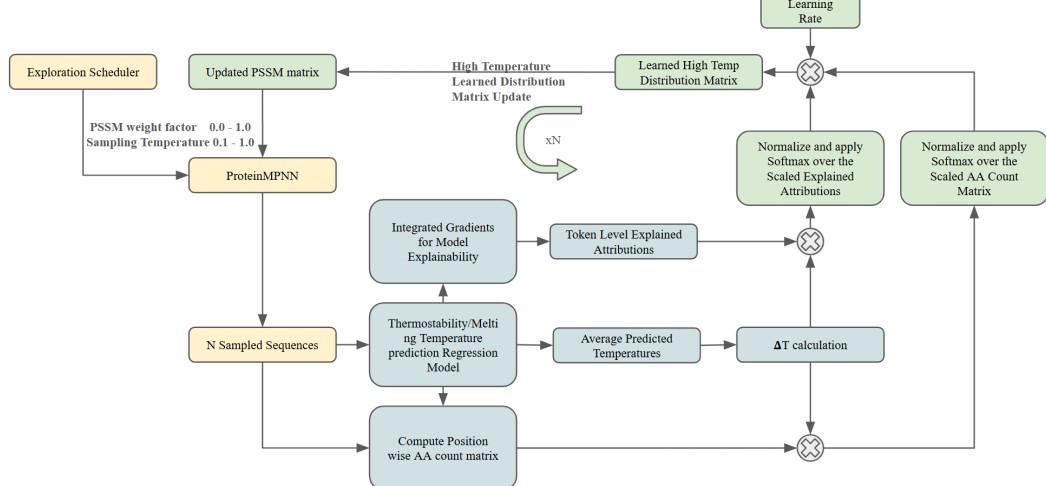

Figure 1: XPro-Design architecture. XPro-Design is an iterative explainable framework that optimizes the PSSM input to bias sampling towards high scoring sequences.The N is a hyperparameter that indicates the number of episodes XPro-Design trains for. Yellow blocks show the sampling process, blue blocks describe the temperature prediction and integrated gradients based attributions calculations while the green colored blocks show the update mechanism.

focus on predicting melting temperatures directly from sequence information, while transformer-based models like TemBERTure (Rodella et al., 2024), ESMStabP (Ramos et al., 2025), TemStaPro (Pudžiuvelytė et al., 2024), and ESMTherm/EsmTemp (Chu et al., 2024; Sułek et al., 2024) build on large pretrained protein models to capture thermostability across diverse families. Structure-aware neural networks, including ThermoMPNN (Dieckhaus et al., 2024), ThermoMPNN-D (Dieckhaus & Kuhlman, 2025), and SPURS (Li & Luo, 2025), further incorporate backbone geometry to refine predictions for thermostability and $T_m$. While these tools provide valuable guidance for engineering, most have been trained primarily on single- or double-point mutations and often show reduced accuracy when generalized to multi-site or structure-wide mutational designs, limiting their utility in large-scale protein re-engineering.

In parallel, generative modeling approaches have shifted the focus from prediction to design. Sequence-to-structure inverse folding methods such as ProteinMPNN (Dauparas et al., 2022), Hy-perMPNN (Ertelt et al., 2024), ESM-IF (Hsu et al., 2022), AlphaDesign (Jendrusch et al., 2025), DivPro (Zhou et al., 2025) and PiFold (Gao et al., 2022) generate sequences compatible with given backbones, and in some cases have been experimentally validated for stable protein design. Diffusion and flow-based frameworks, including RFdiffusion (Watson et al., 2023), MapDiff (Bai et al., 2025), RareFold (Li et al., 2025), and ADFLIP (Yi et al., 2025) introduce probabilistic sampling strategies that enable exploration of novel topologies and controlled backbone design. Specialized models such as AntiFold (Høie et al., 2024) extend generative principles and modeling to antibody design or specialized protein families. However, most of these approaches are trained on broad protein structure datasets (Berman et al., 2000a) and are not optimized for task-specific objectives such as thermostability. For this reason, we adopt ProteinMPNN as a general-purpose baseline and Hy-perMPNN as a task-specific prior for high-temperature stability design, ensuring that comparisons are grounded in models directly relevant to our objective.

XPro-Design enhances inverse folding frameworks such as ProteinMPNN by overcoming biases from mesophile-dominated training data, which undersample rare but functionally important substitutions. By correcting this bias, XPro-Design increases access to underrepresented residues while preserving backbone compatibility and foldability. In addition, it leverages tempering and annealing to broaden exploration of sequence space while refining promising directions, enabling more effective navigation of the mutational landscape. Sequences are sampled in paired batches: one modified by a learned low-rank biasing matrix and the other left unbiased as a control. Each batch is evaluated with melting-temperature (Tm) predictors, wrapped with Integrated Gradients (Sundararajan

et al. (2017)) modules to generate residue-level attribution maps. By aggregating attribution signals across multiple predictors trained on diverse datasets, XPro-Design reduces model-specific biases and builds robust residue-level guidance for updating biasing weights, while the unbiased batch preserves exploration. The framework also supports position masking, allowing functional domains such as catalytic residues or binding pockets to remain fixed while surrounding regions are optimized. In experiments, XPro-Design generated sequences with reduced conservation relative to mesophilic proteins yet achieved substantial improvements in predicted thermostability. In some scaffolds, engineered variants exhibited predicted Tm increases of up to 90 °C compared to wild type. All redesigned variants were validated using Boltz-2 (Passaro et al., 2025) and AlphaFold3-based predictors like Chai-1 (team et al., 2024) and AlphaFold3 (Abramson et al., 2024), which confirmed correct folding into stable structures. Further validation with BioEmu (Lewis et al., 2025) equilibrium sampling, short molecular dynamics simulations (Hollingsworth & Dror, 2018), and MM/GBSA (Sun et al., 2014; Genheden & Ryde, 2015) folding-energy calculations supported favorable $\Delta\Delta G$ changes relative to wild type and baseline generative approaches. While thermostability serves as a case study, XPro-Design establishes a generalizable, explainable, and controllable framework for multi-objective protein engineering, with applications ranging from altered substrate selectivity and enhanced cofactor binding to optimized hinge dynamics and reduced immunogenic epitopes.

## 2 METHODOLOGY

XPro-Design architecture shown in Fig 1 can be broken down into 3 steps which we will deep dive into below.

- Sampling.

- Property prediction and Explaination.

- Update methodology for the PSSM prior for learning the Sequence Preferences.

For the targets, we selected two proteins with distinct thermostability profiles. The first was Candida Antarctica Lipase B (CalB) (Uniprot ID: P41365; (Berman et al., 2000b)), a widely used biocatalyst in esterification, transesterification, and hydrolysis reactions (PDB ID: 4K6G; (Berman et al., 2000b; Xie et al., 2014a)). Wild-type CalB has a reported melting temperature of 45–60 °C ((Xie et al., 2014b; Le et al., 2012; Qian et al., 2009)) depending on variant, solvent and assay conditions, making it moderately stable but suboptimal for high-temperature industrial processes. To preserve activity during optimization, catalytic pocket residues (Ser105, His224, Asp187, and surrounding binding-site residues. Fig. 7) were conserved by leaving them unmasked. The second target was Superoxide Reductase (SOR) (Uniprot ID: P82385) from Pyrococcus Furiosus (PDB ID: 1DQI; (Yeh et al., 2000)), a hyperthermophilic enzyme stable up to 95 °C in oligomeric form ( 75 °C as monomer). SOR was used as a control scaffold to benchmark improvements from XPro-Design against baseline models (ProteinMPNN, HyperMPNN).

### 2.1 SAMPLING

XPro-Design leverages inverse folding models to map three-dimensional protein backbones to sequence space. While we use ProteinMPNN and HyperMPNN in XPro-Design, the framework is general and can incorporate any inverse folding model. Depending on the design objective, critical residues can be preserved by leaving them unmasked during preprocessing; in our experiments, the catalytic and substrate binding residues were explicitly conserved (Fig. 7), though in practice this can extend to entire substrate-binding domains. Given a protein backbone, the inverse folding model generates conditional probabilities over masked positions while respecting frozen residues. We first sample from the baseline model to obtain amino acid distributions, which serve as priors. XPro-Design then tempers this distribution once by applying a temperature, broadening support and reducing initialization bias. Training thereafter proceeds normally, with the tempered distribution gradually sharpening to a newer distribution. This step mitigates the bias of training data dominated by mesophilic proteins, raises entropy, and prevents the model from becoming trapped in local optima early in optimization.

We represent the amino acid distributions across an aligned protein of length $L$ as a matrix

$$P = (p_{i,\ell}) \in \mathbb{R}^{K \times L}, \; K = 20, \quad \sum_{i=1}^{K} p_{i,\ell} = 1 \; \forall \ell, \; p_{i,\ell} \geq 0 \tag{1}$$

To reduce bias toward highly frequent residues (e.g., conserved amino acids) and to increase the chance of sampling low-probability substitutions, we apply temperature scaling independently to each column. For temperature $T > 0$ and smoothing constant $\varepsilon > 0$, the tempered probabilities are defined as

$$\tilde{p}_{i,\ell} = (p_{i,\ell} + \varepsilon)^{1/T}, \qquad p_{i,\ell}(T) = \frac{\tilde{p}_{i,\ell}}{\sum_{j=1}^{K} \tilde{p}_{j,\ell}} \tag{2}$$

**Notation.**

- $K = 20$: number of categories (amino acids).
- $L$: sequence length (positions).
- $p_{i,\ell}$: normalized probability of amino acid $i$ at position $\ell$.
- $\varepsilon > 0$: smoothing constant ensuring nonzero support for all categories.
- $T$: temperature parameter; $T = 1$ recovers the original distribution, $T > 1$ broadens the distribution so that rare amino acids become more likely at a given site, and $0 < T < 1$ sharpens the distribution, reinforcing the dominant residue choices.
- $p_{i,\ell}(T)$: tempered probability of amino acid $i$ at position $\ell$.

After the one-time tempering step, we applied a linear annealing schedule (from T=1.0 to T=0.1) during sampling, progressively sharpening the distribution and enabling early exploration followed by exploitation. Alternatively, sequences can be sampled in parallel from fixed temperatures (e.g., [0.1, 0.5, 1.0]) to balance exploration and exploitation. Sequence generation uses two complementary strategies: direct sampling from the baseline model and sampling guided by a learned position-specific scoring matrix (PSSM) that biases toward desired residue preferences. Together, these yield a diverse baseline of sequences drawn from both the prior and a tempered distribution. A key advantage of this approach is that model weights remain unchanged. This avoids catastrophic forgetting and prevents convergence to narrow local optima, while also preserving the structural fidelity of the inverse folding model; something that can degrade under fine-tuning on limited datasets.

## 2.2 TEMPERATURE PREDICTION AND EXPLAINABLE AI

Each generated protein sequence $x = (x_1, \ldots, x_L), \quad |x| = L$, is evaluated using Integrated Gradients (IG) to attribute residue-level contributions to the predicted melting temperature ($T_m$). While we primarily employ three variants of `TemBERTure` for prediction, the framework can incorporate any differentiable $T_m$ or $\Delta \Delta G$ model. Predictions from `DeepSTABp` are also used to establish cross-model correlations and derive consensus thermostability estimates.

For a differentiable predictor $f_\theta$, the IG for residue $i$ is defined as

$$IG_i(x) = (x_i - x_i') \int_0^1 \frac{\partial f_\theta \big( x' + \alpha(x - x') \big)}{\partial x_i} \, d\alpha \tag{3}$$

where $x_i$ is the embedding vector of residue $i$ in sequence $x$, $x_i'$ is a baseline embedding for residue $i$ (e.g., all-zero or reference amino acid), $\alpha \in [0, 1]$ is the interpolation coefficient along the path from baseline to input, $f_\theta(x)$ is the predicted Tm from model $\theta$ and $IG_i(x)$ is the contribution of residue $i$ to the predicted Tm. Signed attributions $IG_i(x)$ indicate whether a residue increases or decreases predicted $T_m$. Averaging across sequences helps suppress predictor noise, yielding sharper and more reliable attribution signals.

Pairwise effects can be captured by

$$IG_{ij}^{\text{pair}} = \frac{1}{|S|} \sum_{x \in S} IG_i(x) \cdot IG_j(x) \tag{4}$$

where positive values denote synergistic contributions and negative values denote antagonistic interactions. Although this signal is not directly used to train XPro-Design, this matrix is a functional epistasis map and is very useful in monitoring how the model believes amino acid positions interact to affect the prediction. It also reveals residue–residue synergy or antagonism across the optimization loop

## 2.3 LEARNING SEQUENCE PREFERENCES

Each generated batch of sequences is converted into a position-specific scoring matrix (PSSM) of shape $L \times 20$, where $L$ is the sequence length and 20 corresponds to the amino acid types. To update the PSSM, we first compute the mean-centered stability signal for each sequence:

$$\Delta T_m(x) = T_m(x) - \mu_{T_m} \tag{5}$$

where $T_m(x)$ is the predicted melting temperature of sequence $x$, and $\mu_{T_m}$ is the batch mean.

To enhance signal separation, $\Delta T_m$ values are rescaled on an exponential scale. For a batch of $\Delta T_m$ values, let

$$\Delta T_m^{\max} = \max_{x \in S} |\Delta T_m(x)| \tag{6}$$

We then define

$$\widetilde{\Delta T}_m(x) = \Delta T_m(x) \cdot \left( \frac{|\Delta T_m(x)|}{\Delta T_m^{\max}} \right)^{\gamma} \tag{7}$$

where $\gamma > 0$ is an exponent hyperparameter.

This transformation ensures that large deviations from the mean are amplified, small deviations are attenuated and the sign of $\Delta T_m(x)$ is preserved.

The weighted $\Delta T_m$ is then combined with residue-wise attribution scores to form the update term:

$$\Delta \text{PSSM}_{i,a} = \frac{1}{|S|} \sum_{x \in S} \widetilde{\Delta T}_m(x) \cdot IG_{i,a}(x) \tag{8}$$

where $IG_{i,a}(x)$ is the attribution score for amino acid $a$ at position $i$ in sequence $x$, and $S$ is the batch of sequences.

This procedure ensures that amino acids contributing to low $T_m$ are penalized with amplified negative updates, amino acids contributing to high $T_m$ are rewarded with amplified positive updates, context-dependent effects (epistasis) are captured naturally through batch-level averaging.

Finally, the PSSM is updated iteratively as

$$\text{PSSM}^{(t+1)} = \text{PSSM}^{(t)} + \eta(t) \cdot \Delta \text{PSSM} \tag{9}$$

where the learning rate $\eta(t)$ follows a linear decay schedule from 0.01 to 0.001. Over successive batches, this update biases sampling toward regions enriched in stabilizing mutations while maintaining exploration of diverse sequence space.

## 2.4 EVALUATION OF FOLD INTEGRITY AND STABILITY

After convergence of the sampling distribution, we generated N sequences for evaluation. Thermostability was first predicted using the methods in Section 2.2, followed by structure prediction with Boltz-2. Predicted structures were aligned to the reference backbone, and C$\alpha$ RMSD was calculated to assess fold preservation. Additional descriptors like packing density, solvent-accessible surface area (SASA), and inter-residue interaction networks—were computed on both Boltz-2 and energy-relaxed structures to evaluate packing and interaction integrity. For thermodynamic validation, sequences were analyzed with BioEmu to sample 50 equilibrium conformations, subjected to molecular dynamics simulations, and evaluated by MM/GBSA free-energy calculations in OpenMM ((Eastman et al., 2017)). These MM/GBSA energies (denoted as $\Delta G$ in tables and plots) are used as relative stability proxies of folded conformations; they should not be interpreted as absolute folding free energies ($\Delta G_{\text{fold}}$). The resulting $\Delta \Delta G$ values quantified relative stability across variants. This integrated framework ensured that designed proteins preserved structural topology while exhibiting favorable energetic and thermodynamic profiles.

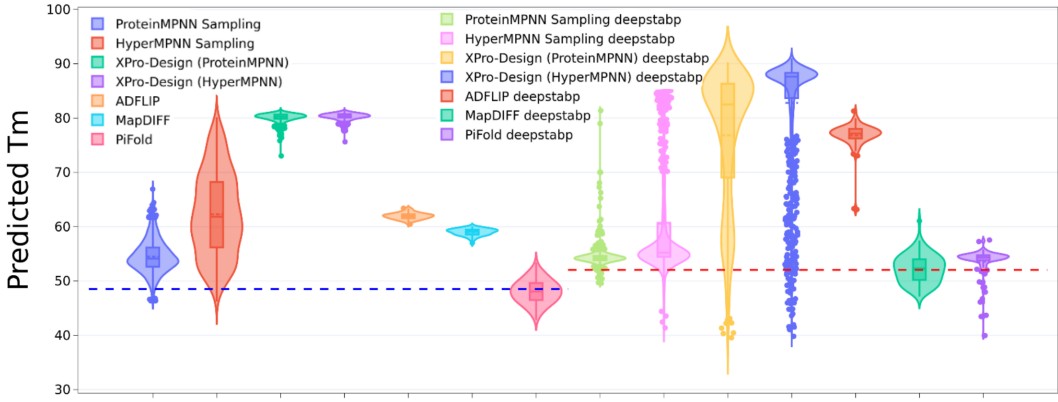

Figure 2: Predicted melting temperature values for CalB variants in °C for the different methods using TemBERTure (left) and DeepSTABp (right) models. The dashed lines are the wildtype protein predicted temperatures for the 2 prediction models used (TemBERTure and DeepSTABp). We Observe the spread of High-Tm sequences is considerably narrower for XPro-Design compared to even HyperMPNN which was specifically trained to generate thermophile variants.

## 3 RESULTS

### 3.1 MELTING TEMPERATURE PREDICTIONS

Once XPro-Design had learned the optimized protein sequence space, we generated 1,000 sequences each from ProteinMPNN, HyperMPNN, ADFLIP, MapDIFF, PiFold, XPro-Design(P) (with Protein-MPNN sampling), and XPro-Design(H) (with HyperMPNN sampling) at sampling temperatures of 0.1 and 0.5. Key point to note here is that the models ADFLIP and MAPDIFF do not have a sampling temperature parameter, this results in a very low diversity score for both the methods since these tend to generate sequences with very high sequence recovery to the WT protein leading to very few mutations and low diversity within that mutational space as well.

As illustrated in Figure 2, our approach outperformed even the specialized HyperMPNN model in generating sequences with substantially higher predicted melting temperatures (Tm). Thermostability was independently verified using both TemBERTure and DeepSTABp predictors, with Deep-STABp evaluated at a growth temperature of 37°C. Consistent with our hypothesis, XPro-Design reliably identified high-Tm sequence variants for the given backbones regardless of the underlying sampling model. The difference between XPro-Design(P) and (H) was negligible in terms of predicted temperature performance and sampled AA space, indicating strong generalization.

Among the baseline methods, ProteinMPNN, MapDIFF and PiFold exhibited the weakest performance, with the lowest mean and maximum predicted $T_m$ across both TemBERTure and Deep-STABp (Table 1). HyperMPNN generated sequences with higher maximum temperatures than ProteinMPNN, though mean values remained significantly below those achieved by our methods.

In contrast, both XPro-Design variants consistently produced sequences with markedly elevated thermostability. TemBERTure predicted mean $T_m$ values above 80 °C for both samplers, while DeepSTABp predictions reached up to 91 °C. The temperature distribution was slightly sharper for XPro-Design(H) as the sampler, though the overall improvement over XPro-Design(P) was marginal when considering sequence recovery and diversity as shown in Table 1.

### 3.2 SEQUENCE DIVERSITY AND RECOVERY

Sequence diversity was quantified using a k-mer–based Jaccard similarity approach ($k = 3$), which efficiently captures local compositional differences without requiring full pairwise alignments. Diversity is expressed as 1 minus the Jaccard similarity (Brohee & Van Helden, 2006), averaged over

Table 1: Predicted Temperatures and Sequence Recovery.

| Methods | TemBERTure Tm (°C) mean, max ↑ | DeepSTABp Tm (°C) mean, max ↑ | Sequence Recovery | Sequence Diversity |
|---|---|---|---|---|
| ProteinMPNN | 54.4, 66.9 | 54.4, 81.3 | 56.3% ± 2.5% | 0.750 |
| HyperMPNN | 62.2, 80.1 | 60.5, 90.0 | 48.7% ± 1.5% | 0.732 |
| ADFLIP | 61.9, 63.3 | 76.8, 81.2 | 68.7% ± 0.1% | 0.061 |
| MAPDIFF | 58.9, 60.1 | 52.1, 61.1 | 83.0% ± 0.6% | 0.201 |
| PiFold | 48.0, 53.3 | 53.5, 57.5 | 65.9% ± 3.3% | 0.755 |
| XPro-Design(P) ours | **80.1, 81.5** | **76.8, 90.2** | 48.2% ± 1.5% | 0.734 |
| XPro-Design(H) ours | **80.3, 81.5** | **82.7, 90.9** | 35.9% ± 1.4% | 0.602 |

Table 2: Structure Predictions

| methods | Boltz-2 PTM | Boltz-2 pLDDT | RMSD (Å) ↓ |
|---|---|---|---|
| ProteinMPNN | 0.966 ±0.004 | 0.94 ±0.01 | 1.94±0.25 |
| HyperMPNN | 0.958 ±0.007 | 0.92 ±0.01 | 2.02 ±0.33 |
| ADFLIP | 0.964 ±0.001 | 0.93 ±0.01 | 3.12 ±0.16 |
| MAPDIFF | 0.969 ±0.001 | 0.94 ±0 | 2.94 ±0.1 |
| PiFold | 0.963 ±0.006 | 0.93 ±0.01 | 3.1 ±0.33 |
| XPro-Design (P) ours | 0.947 ±0.009 | 0.90 ±0.01 | 2.24 ±0.69 |
| XPro-Design (H) ours | 0.915 ±0.015 | 0.85 ±0.02 | 3.12 ±0.56 |

all sequence pairs. As shown in Table 1, all variants exhibit comparable diversity except XPro-Design(H), which converged on a narrower sequence space.

Sequence recovery was computed as the fraction of residues matching the wild-type (WT) sequence at aligned positions, reflecting the balance between conservation and exploration. MapDIFF and ADFLIP achieved the highest recovery (83.0% and 68.7%), indicating strong preservation of WT residues but severely limited exploration, which results in low diversity, weaker thermostability and poor $\Delta\Delta G$ improvements. HyperMPNN (48.7% recovery) explores more of sequence space, yielding more thermophilic designs, though gains remain modest. It is important to note here that for optimization tasks like thermostability, high sequence recovery is not ideal since we need to sample from the hyper-thermophile space which is quite different from the mesophilic proteins. On the contrary, models generating sequences with very high sequence recovery are essentially sampling in the mesophilic space and do not generate enough diversity to explore thermophilic proteins (as evidence by their low pTm, stability scores and poor inter-residue interaction profiles). Hence it is important to strike a good balance between sequence recovery/conservation and diversity to explore the thermophilic design space effectively.

XPro-Design variants achieved recoveries of 48.2% (P) and 34.7% (H) while generating sequences with markedly improved thermostability and reduced $\Delta\Delta G$. Lower recovery for HyperMPNN-based designs reflects its bias toward hyperthermophilic residues which is very different from mesophilic WT protein. XPro-Design(P) strikes an optimal balance, maintaining high sequence coverage while producing superior designs without model-specific fine-tuning. Both XPro-Design variants converge to a distinct amino acid distribution, clearly separating them from the baseline models while exploring very diverse areas of hyper-thermophilic protein space.

## 3.3 STRUCTURE PREDICTION AND RMSD

We predicted the structures for all generated sequences using the Boltz-2 model, with multiple sequence alignments (MSAs) obtained from the ColabFold (Mirdita et al., 2022) MSA server. Wild-type templates were not provided during the prediction to ensure unbiased folding assessments. Across all methods, the designed sequences were predicted to fold correctly, with XPro-Design variants consistently demonstrated successful folding. Backbone RMSD values of the designed sequences relative to the wild-type backbone showed minimal deviations, confirming structural conservation despite extensive sequence redesign. As illustrated in Figure 6, the variant V_2372 folded

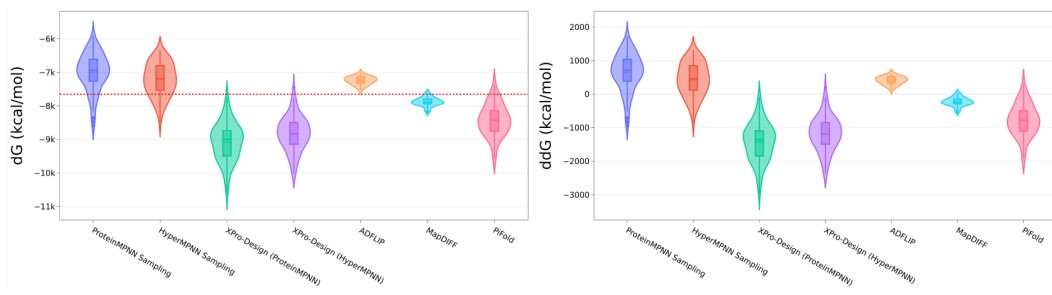

Figure 3: Violin plots showing $\Delta G$ (left) and $\Delta\Delta G$ (right) distributions of variants generated from the different methods computed against CalB WT.

Table 3: Folding free energy, $\Delta\Delta G$ and

| Methods | $\Delta G$ mean, best (kcal $\cdot$ mol$^{-1}$) ↓ | $\Delta\Delta G$ mean, best ($\pm$ kcal$\cdot$mol$^{-1}$) ↓ | Normalized entropy (J$\cdot$mol$^{-1}\cdot$K$^{-1}$) ↓ | Normalized entropy Core (J$\cdot$mol$^{-1}\cdot$K$^{-1}$) ↓ |
|---|---|---|---|---|
| ProteinMPNN | -6999, -8588 | 650, -939 | 0.707 ± 0.010 | 0.696 ± 0.013 |
| HyperMPNN | -7215, -8504 | 433, -855 | 0.682 ± 0.009 | 0.681 ± 0.011 |
| ADFLIP | -7242, -7526 | 407, 123 | 0.751 ± 0.002 | 0.698 ± 0.002 |
| MAPDIFF | -7875, -8176 | -226, -527 | 0.706 ± 0.002 | 0.678 ± 0.003 |
| PiFold | -8423, -9563 | -774, -1913 | 0.720 ± 0.005 | 0.690 ± 0.004 |
| XPro-Design(P) | **-9075,-10612** | **-1426, -2962** | **0.673 ± 0.009** | **0.670 ± 0.012** |
| XPro-Design(H) | **-8828, -10026** | **-1179, -2377** | **0.660 ± 0.007** | **0.658 ± 0.010** |

nearly identically to the wild-type CalB backbone while exhibiting a 38.7% reduction in predicted $\Delta\Delta G$ and a 63% improvement in predicted melting temperature. Predicted Template Modelling score (PTM) and predicted Local Distance Difference Test (pLDDT) scores are summarized in Table 2, all within acceptable confidence thresholds, further supporting the structural reliability of the generated sequences.

## 3.4 FREE ENERGY AND PACKING ENTROPY ANALYSES

We evaluated thermodynamic stability using folding free energies ($\Delta G$) and relative stability changes ($\Delta\Delta G$) with respect to the wild type (Table 3). Stabilizing variants were defined by lower $\Delta G$ and negative $\Delta\Delta G$. Baseline ProteinMPNN generated stabilizing variants in only 14% of cases, indicating overall destabilization. HyperMPNN modestly improved performance (21% stabilizing). ADFLIP generated 0% stabilizing variants, while PiFold and MapDIFF achieved the better results, producing 44% and 46% stabilizing variants, respectively. These results confirm that baseline models rarely introduce consistently stabilizing substitutions.

In contrast, XPro-Design produced near-universal stabilization (Fig. 3). With ProteinMPNN as the sampler, all variants were stabilizing, with mean $\Delta G$ = –9075 kcal·mol$^{-1}$ and mean $\Delta\Delta G$ = –1426 kcal·mol$^{-1}$. Using HyperMPNN yielded similarly strong results (99% stabilizing, mean $\Delta G$ = –8828 kcal·mol$^{-1}$, mean $\Delta\Delta G$ = –1179 kcal·mol$^{-1}$). Both samplers achieved substantially more favorable $\Delta\Delta G$ values than either baseline, consistent with the enhanced thermostability and packing analyses (Fig. 4).

To further probe stability, we computed packing-derived residue entropies with PACKMAN (Khade, 2024) (Voronoi/Delaunay geometry → packing fraction → entropy). Normalized entropy values (removing length bias)(Fig 11) showed clear reductions for XPro-Design. Hydrophobic-core entropies followed the same trend (Table 3), confirming that XPro-Design variants adopt tighter, less flexible packing. These paired results indicate that improved thermostability arises from redistributed packing patterns rather than simple global compression of the core volume.

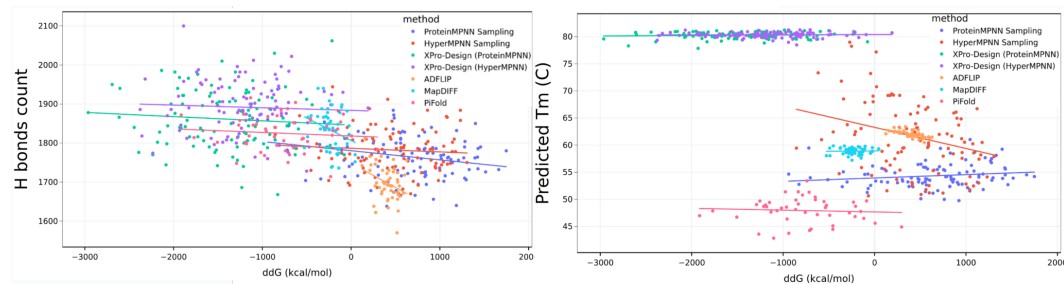

Figure 4: Scatter Plot of H-bond count (left) and Predicted melting Temperatures (right) vs ddG. The lines show the fitted linear regiression line for the different methods. The plots show that XPro-Design generates denser h-bond networks within the protein resulting in improved stability and melting temperatures.

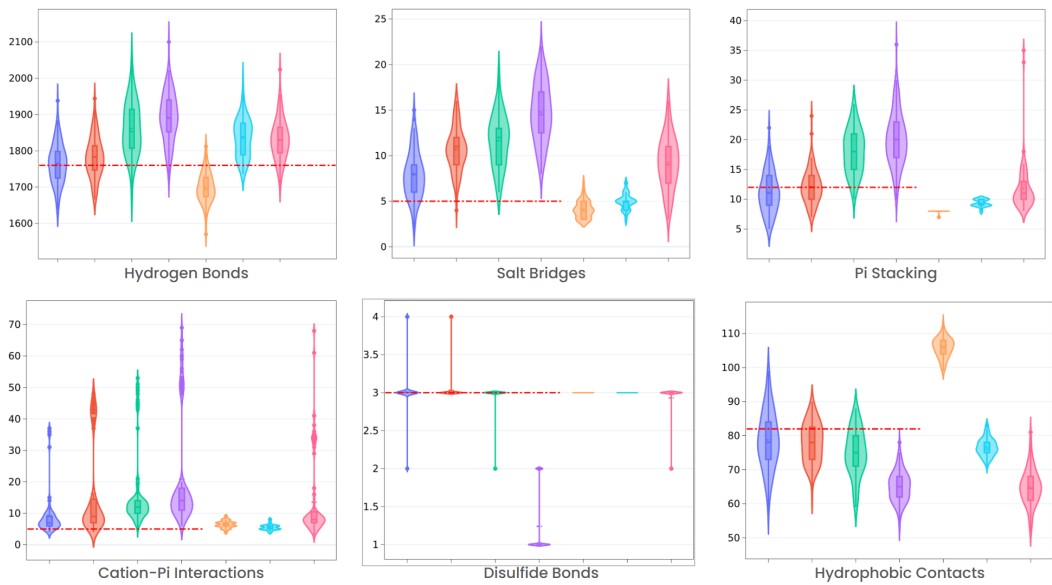

Figure 5: Inter residue Interactions: ProteinMPNN (blue), HyperMPNN (red), XPro-Design(P) (green), XPro-Design(H) (purple), ADFLIP(orange), MAPDIFF(light-blue), PiFold(pink).

## 3.5 BOND ANALYSIS

The analysis of non-covalent interactions revealed that XPro-Design variants consistently formed a higher number of hydrogen bonds relative to other methods (Fig. 5). In the same figure, a comparable trend was observed for salt bridges, $\pi$-cation interactions, and $\pi$-stacking contacts, all of which increased significantly in the redesigned variants. Disulfide bond counts remained largely unchanged across methods, indicating that the global covalent connectivity of the proteins was preserved.

Interestingly, this redistribution of interactions was accompanied by a modest reduction in hydrophobic contacts. However, rather than reflecting destabilization, this shift appears to be a compensatory effect: the gain in directional, energetically favorable interactions such as hydrogen bonds and electrostatic or aromatic contacts outweighs the slight decrease in non-specific hydrophobic packing. This is further demonstrated in Fig. 9 and Fig. 10.

Taken together, these results indicate that XPro-Design variants achieve improved stability not by maximizing hydrophobic burial alone, but by reinforcing a diverse network of stabilizing non-covalent interactions. This richer interaction landscape likely contributes to the enhanced thermostability and folding robustness observed in our designs.

## 4 CONCLUSION

We introduced XPro-Design, a novel framework for protein sequence optimization that leverages explainable AI in a gradient-free setting. By using attribution methods such as Integrated Gradients, the approach provides residue-level interpretability while guiding optimization with task-specific predictors. Unlike conventional baselines, XPro-Design requires no fine-tuning of the model weights, operating directly on inverse folding models while retaining their structural fidelity and broad applicability. The framework balances exploration and exploitation through tempered initialization and annealed sampling, systematically uncovering stabilizing mutations missed by baseline methods. As a result, XPro-Design designed orders of magnitude more stable sequences than other SOTA methods. It consistently yielded higher predicted thermostability, near-universal shifts toward stabilizing $\Delta\Delta G$, improved inter-residue interaction networks and reduced packing entropies indicative of more favorable folds; all while preserving sequence diversity. These results highlight XPro-Design as a transformative step in protein engineering: scalable, interpretable, and capable of delivering unprecedented improvements in stability beyond the reach of existing generative models.

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

## A  APPENDIX

### A.1  FUTURE WORK

Future work planned around XPro-Design involves validating some of the top variants in a lab along with quantifying their half lives at an elevated range of temperatures compared to the WT proteins. Also we are currently working on redesigning several other industry relevant proteins to operate at elevated temperatures.

Moreover we have already started testing XPro-Design towards substrate binding selectivity for enzymes, optimizing protein-protein binding affinity by improving the interaction profiles on the binding domains as well as evaluating XPro-Design towards improving enzymatic kinetics by redesigning hinge regions towards higher catalytic efficiency.

Lastly, we are working on permissions to open-source the entire code base as well.

### A.2  SHANNON ENTROPY

The Shannon entropy of a categorical distribution $p$ is defined as

$$H(p) = -\sum_{i=1}^{K} p_i \log p_i. \tag{10}$$

Applying tempering yields the distribution $p(T)$, whose entropy is

$$H(p(T)) = -\sum_{i=1}^{K} p_i(T) \log p_i(T) \tag{11}$$

For $T > 1$, it follows that

$$H(p(T)) \geq H(p),$$

with equality if and only if $p$ is uniform.

**Notation:**

- $H(p)$: Shannon entropy of distribution $p$.
- log: natural logarithm.
- $p_i(T)$: tempered probability of category $i$.

### A.3 Biological intuition behind Temperature Scaling of prior distribution

Temperature scaling with $T > 1$ increases the probability of rare substitutions that might otherwise be ignored because of under representation in the training data, thus encouraging exploration of sequence diversity. Conversely, setting $T < 1$ amplifies the dominance of conserved residues, reinforcing evolutionary constraints. This single parameter therefore provides a biologically interpretable knob to balance between conservation and diversity in sampling.

### A.4 CalB Structural Results

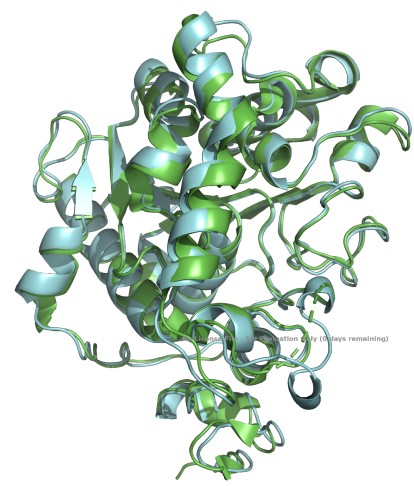

Figure 6: lowest ddG variant V_2372 (blue) overlaid over WT 4K6G (green) structure

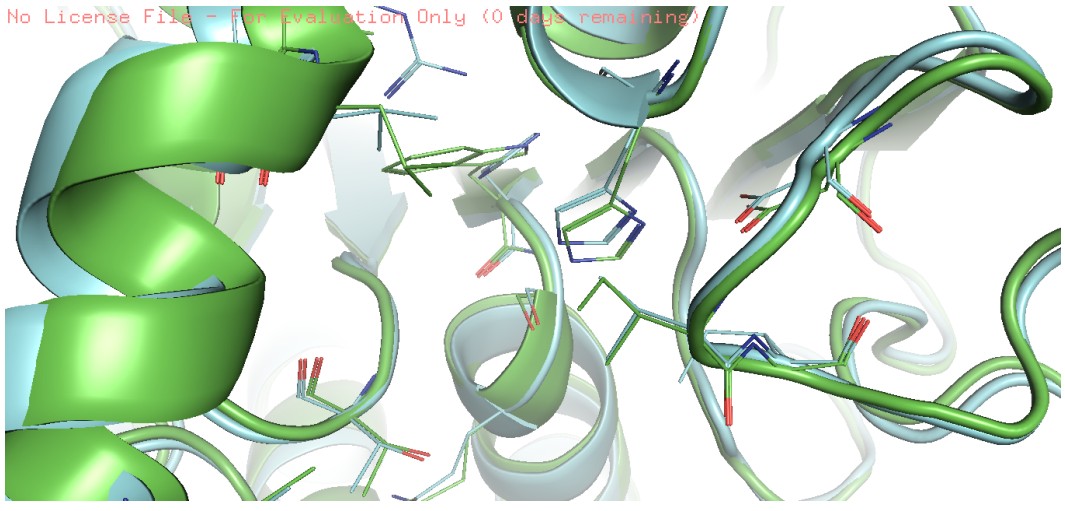

Figure 7: lowest ddG variant V_2372 (blue) overlaid over WT 4K6G (green) structure shows that the substrate bidning pocket is well conserved. Catalytic Triad Residues S105, D187, H224 and Substrate binding residues T40, E188, L278 were specifically conserved. Residues I189, V190, I285 were masked out, yet we see that XPro-Design substitutions represents a conservative change (Ile → Leu) within the substrate binding pocket, and is not expected to drastically alter the overall hydrophobic character even for redesigned residues not explicitly conserved, though subtle changes in side-chain packing or pocket geometry may occur.

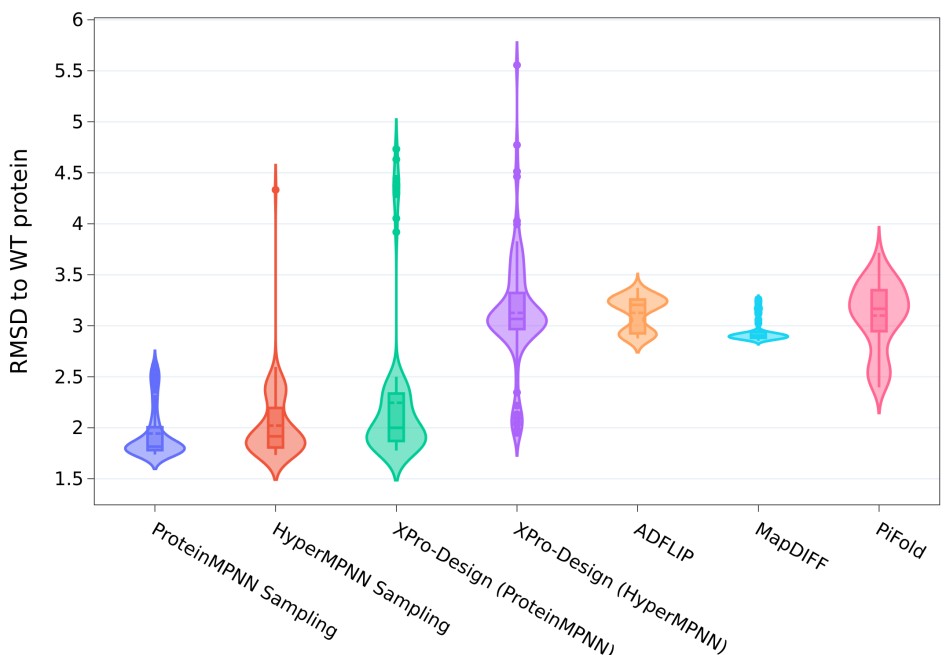

Figure 8: Predicted structure backbone RMSD from WT CalB.

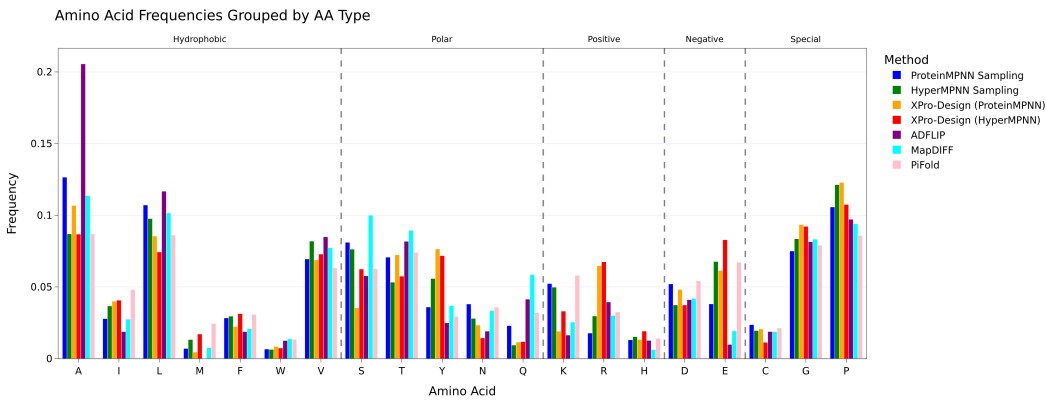

Figure 9: AA wise distribution shift of XPro-Design from baseline models

## A.5 AA-WISE DISTRIBUTION SHIFTS

To evaluate how design strategies altered amino acid usage, we analyzed the distribution of residue classes across surface, core, and overall regions of the protein (Figure 10 & 9). Both baseline samplers (ProteinMPNN and HyperMPNN) preserved broad compositional trends but differed in their bias toward polar residues on the surface and hydrophobic residues in the core.

XPro-Design introduced a clear shift in these distributions. On the surface, it reduced excessive polar enrichment while slightly increasing charged and special residues, suggesting more balanced solvent exposure. In the core, XPro-Design produced a higher fraction of hydrophobic residues and a modest

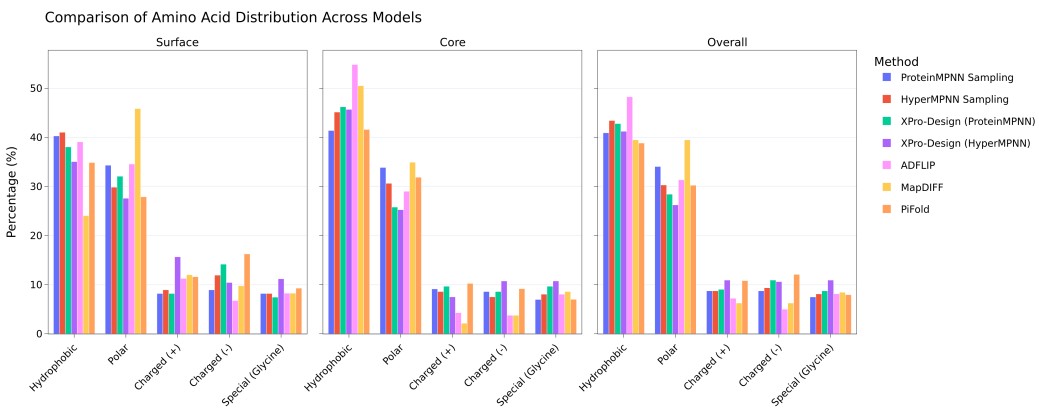

Figure 10: Surface vs Core AA Type distribution from the different methods

rise in glycines, consistent with tighter packing and increased conformational adaptability. When averaged over the full sequence, the distributions from both XPro-Design variants diverged from the baselines in a consistent manner, indicating that optimization not only improved thermostability but also drove distinct residue-level preferences aligned with thermophilic design principles.

## A.6 ENTROPY REDUCTION

We can see that XPro-Design consistently generates variants that have lower core entropy compared to the other methods. This combined with ddG and denser bond networks shows that XPro-Design consistently generates more thermo-stable variants compared to other methods.

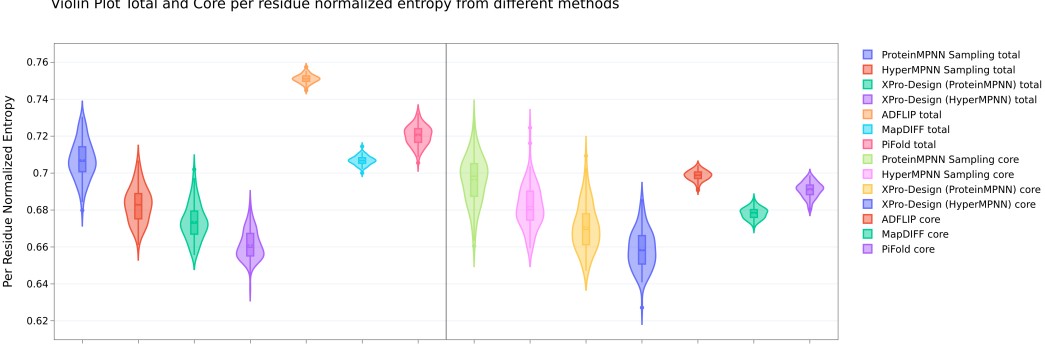

Figure 11: Normalized Per Residue wise Entropy for the full protein (left) and the core (right).

## A.7 SORA RESULTS

Despite the SorA protein being a small protein having only 124 AAs with not much scope towards optimization since it is already a hyper thermophile, we observe a clear and similar trend here as well. Our XPro-Design method considerably outperforms even the finetuned HyperMPNN model at designing more thermostable variants. This is clear by the clear upward shift in the predicted melting temperatures from different methods as well as the MM/GBSA based energy calculations.

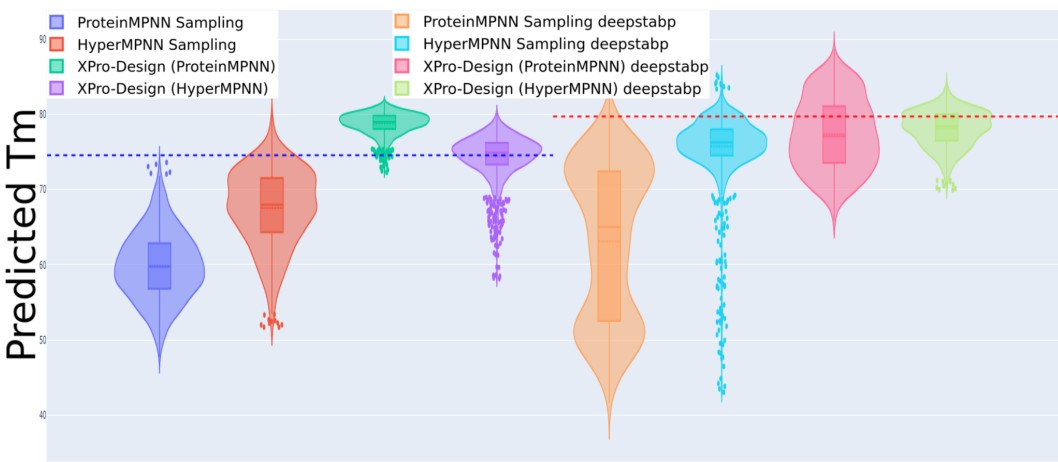

Figure 12: Predicted melting temperature values for SorA variants in °C for the different methods using TemBERTure and DeepSTABp models. We Observe the spread of High-Tm sequences is considerably narrower for XPro-Design compared to even HyperMPNN which was specifically trained to generate thermophile variants.

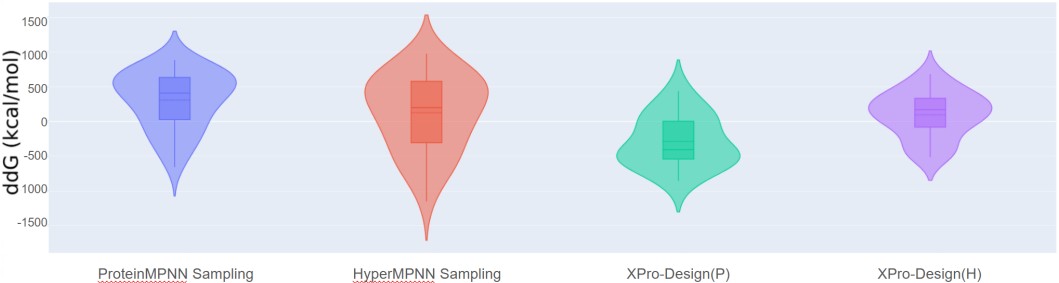

Figure 13: Violin plots showing $\Delta\Delta G$ distributions of variants generated from the different methods computed against SorA WT

Similar to trends seen with the CalB target, all the generated variants from all the methods folded correctly despite having sequence coverage of only around 53% across all methods.

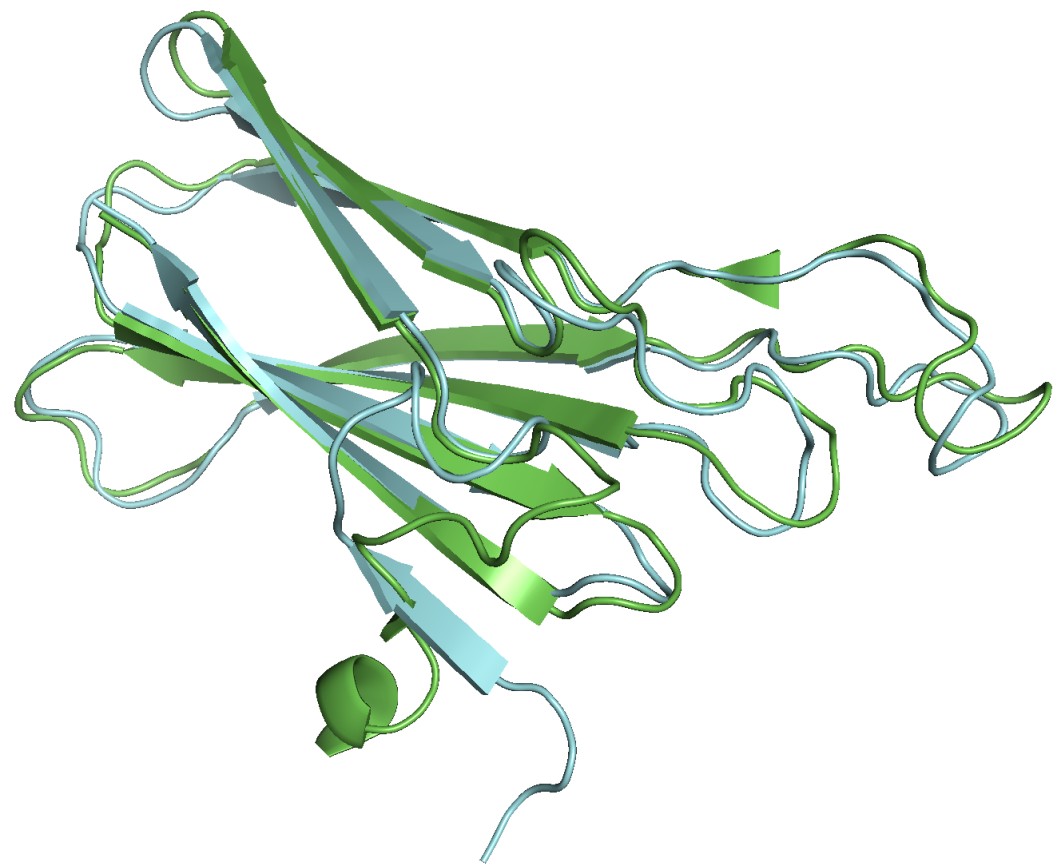

Figure 14: Top SorA designed variant V_5984 (Xpro-Design(P) in Blue overlaid over SorA WT 1DQI

## A.8   LLM USE DISCLAIMER

The authors used a large language model (ChatGPT, OpenAI) to assist in polishing grammar and improving conciseness of the manuscript text. The model was not used for data analysis, generation of scientific content, or drawing conclusions. All scientific content and interpretations are solely the responsibility of the authors.

