# OpenReview forum: "XPro-Design: Rational Protein Engineering Framework Using Explainable AI"
_ICLR.cc/2026/Conference — Submitted to ICLR 2026_

### Official Review · Reviewer_feGP · 2025-10-30

**Soundness:** 1
**Presentation:** 1
**Contribution:** 1
**Rating:** 0
**Confidence:** 3

**Summary:**

The authors claim to contribute:
- a method to shift a pre-trained generative model toward some function of interest with a low-rank (?) per-position scoring matrix
- designed proteins for increased thermostability; method is extensible to other objectives
- tries to address bias (present in most pre-trained models) toward mesophilic proteins by increasing the default temperature of pre-trained models, because they want to design proteins for extreme temperatures.

**Strengths:**

- Results of applying the PSSM to pre-trained inverse folding models leads to an appreciable increase in the distribution mean PTM (Fig. 2).
- Refolding results (Table 2) suggest that protein sequences generated by their procedure fold near the target structure (though not under 2 Angstroms) and that AlphaFold is pretty confident about them.
- Authors try to validate their method in a variety of ways, including some packing analyses.
- Doesn’t seem to require training anything, just combines a pre-trained generative model with a PSSM computed from $f(x)$.

**Weaknesses:**

- I’m confused as to why the authors primarily used interpretable gradients since for protein design, the properties one tries to maximize are not generally differentiable because they’re defined as $f(x)$ with respect to discrete $x$ (protein sequence). One could decide to take gradients of discrete functions if they exist, as they do for neural networks [1], but that should be explained and justified. Moreover, it’s odd that in the explicitly non-differentiable predictor case, they use correlations. One might more naturally consider fitting an epistatic linear model up to order two (position-wise and pairs). An explanation of how integrated gradients and correlations used correspond to rigorous notions of epistasis [2] is lacking.
- Authors explain how to get PSSM given integrated gradients but not for correlations. Also, the PSSM seems to only use the single site IG, and not any pairs — so the claim that the method accounts for epistasis seems inappropriate.
- Authors say PSSM is learned, but I don’t see anywhere describing learning or training. They also say that it’s low-rank, but I don’t see anything about that either.
- It seems odd to use AlphaFold-based refolding metrics (Table 2) if they’re trying to design proteins for extreme conditions; should AlphaFold do a good job on these out-of-distribution proteins?
- Results for only one protein are shown in the main text, making it difficult to evaluate whether this works generally. Their claims about their method seem too strong for the evidence presented. It’s also not evaluated against any of the methods mentioned in the intro, just the base pre-trained models.
- It’s difficult to understand novelty of method because the explanation of the method isn’t well situated within existing methods. It seems to sort of come out of nowhere. The idea seems really simple, which isn’t necessarily a bad thing, but makes me wonder if some other paper has already done something similar, and if this method really achieves SOTA performance for taking a pre-trained generative model and shifting its distribution according to some desired property $f(x)$. There are many generic methods for doing this like fine-tuning, RL post-training, SMC, etc. that aren’t mentioned.
- Organization and clarity of the writing is significantly lacking:
    - No separate related works
    - Figure 1 doesn’t have a caption and is pretty complicated. Fonts in Figure 2 are oddly sized, and the dashed lines aren’t explained in the legend or in the caption. In general, captions aren’t very descriptive or helpful.
    - Grammar hasn’t been thoroughly checked throughout and there’s missing capitalization (e.g., Table 1), typos, etc.
    - The methodology goes straight into experimental details without providing any real roadmap
    - Writing is overly specific in places (e.g., why is a particular length, $L=320$, specified for the first equation, which one would expect to be generic?)
    - Equations aren’t written carefully (e.g., in Eq.2, $\tilde{p}_{i,l}$ should take $T$ as an argument)
    - Text and figures aren’t consistent; e.g., Sec. 3.3 says that Table 2 shows PAE but it does not.

I recommend rejecting this paper for two primary reasons — their idea might be good, but there isn’t sufficient evidence of it working beyond one test protein, and the paper overall is not well written. I think substantial revision effort is needed to make this paper understandable.

A few small pieces of advice: It would be clearer to the reader if you had a separate related works section, as opposed to interspersing it throughout your introduction. Correspondingly, it would also be easier to understand your paper if the introduction primarily focused on the motivation for your method and its big picture idea with intuition. As it stands, the introduction is a bit scattered and claims that your method will do many things without making it clear to the reader what the (at most 3) critical contributions you make are.
There’s also a red squiggly typo line in your Figure 1. An algorithm box, or at least a clear outline somewhere, would help the reader to understand what exactly your general method is, separate from the specific analyses that you do throughout. And please, please, structure your paper (and especially your methods section) clearly, as if you're trying to teach someone what you did.

[1] Grathwohl, Will, et al. "Oops i took a gradient: Scalable sampling for discrete distributions." International Conference on Machine Learning. PMLR, 2021.
[2] Poelwijk, Frank J., Vinod Krishna, and Rama Ranganathan. "The context-dependence of mutations: a linkage of formalisms." _PLoS computational biology_ 12.6 (2016): e1004771.

**Questions:**

The details of the method are overall pretty unclear, I'd like clarity on that primarily. More explanation of how your method is different from the closest related works would help to position it in the field. Comparison with methods other than just pre-trained model would also be informative so readers can understand if your method is achieving SOTA, or how close if not. Comparisons on more proteins or properties other than thermostability would be compelling if this method really is applicable generally.

---

### Official Review · Reviewer_5WzG · 2025-10-30

**Soundness:** 3
**Presentation:** 3
**Contribution:** 3
**Rating:** 4
**Confidence:** 3

**Summary:**

This work proposes the XPro-Design framework, which combines generative inverse folding models with explainable AI to rationally engineer proteins for improved thermostability and other functional objectives. It begins by sampling amino acid sequences from backbone-conditioned models such as ProteinMPNN or HyperMPNN and applies temperature tempering to broaden sequence diversity. Each generated sequence is then evaluated using differentiable thermostability predictors, where Integrated Gradients quantify the residue-level contributions to melting temperature. These attribution scores guide iterative updates of a position-specific scoring matrix (PSSM), biasing future sampling toward stabilizing mutations while maintaining structural fidelity. Through this explainable optimization process, XPro-Design efficiently explores the mutational landscape, refines favorable regions of sequence space, and produces protein variants with enhanced folding stability and preserved functional domains.

**Strengths:**

1 Compared with previous generative models such as ProteinMPNN or HyperMPNN, XPro‑Design incorporates explainable AI methods like Integrated Gradients to identify the contribution of each residue to target properties. This not only improves optimization efficiency but also allows researchers to understand why specific mutations enhance stability, enabling truly rational design rather than black‑box generation.

2 The method employs temperature‑controlled sampling and annealing to balance exploration and exploitation in sequence space. Instead of directly finetuning model parameters, XPro-Design updates a biasing matrix and PSSM iteratively, achieving distribution‑level optimization without catastrophic forgetting or overfitting.

3 The framework demonstrates robust performance across different test templates (e.g., CalB and SOR), generating variants with significantly improved melting temperatures, free energy profiles, and packing entropies. Structural validation using Boltz‑2 and AlphaFold3 confirms correct folding, proving that XPro-Design can generalize effectively and support multi‑objective protein optimization tasks.

**Weaknesses:**

1 XPro-Design employs low‑rank bias matrices and Integrated Gradients to guide explainable optimization. While this provides residue‑level interpretability, the framework relies entirely on gradient‑based signals from thermostability predictors, which may carry noise or bias from models such as TemBERTure or DeepSTABp. This could lead to unstable optimization directions. Please explain why you didn’t incorporate more robust attribution approaches (e.g., SHAP or causal inference) to reduce susceptibility to gradient noises, and add relative baselines.

2 The study mainly compares results with ProteinMPNN and HyperMPNN, which can be a bit simple and make it hard to situate the model’s performance relative to broader state‑of‑the‑art design frameworks. Please include more baselines such as RFdiffusion, MapDiff, ADFLIP.

3 The current metrics may not be enough for comprehensive evaluations. The author may expand the experimental validation dataset or integrate existing high‑throughput stability databases like FireProtDB; use broader biophysical measurements (e.g., unfolding rate, ΔH, or kinetic assays) to support multi‑dimensional validation; and introduce uncertainty quantification metrics (e.g., variance of attribution scores) to assess the reliability of explainability signals.

**Questions:**

Same as weakness.

---

> ### Author Response · Authors · 2025-12-02
>
> **We thank the reviewers for their careful evaluation and constructive feedback. Below, we address each comment in detail.**
>
> **Q1**. XPro-Design employs low‑rank bias matrices and Integrated Gradients to guide explainable optimization. While this provides residue‑level interpretability, the framework relies entirely on gradient‑based signals from thermostability predictors, which may carry noise or bias from models such as TemBERTure or DeepSTABp. This could lead to unstable optimization directions. Please explain why you didn’t incorporate more robust attribution approaches (e.g., SHAP or causal inference) to reduce susceptibility to gradient noises, and add relative baselines.
>
> **Answer:**
> * Thank you, this is a very good question.
> * Section 2.2 already explains why gradient-based attribution is sufficiently stable in our setting. The optimization loop operates on batch-averaged attribution scores, computed over 50–100 sampled sequences per iteration. This averaging substantially suppresses predictor-level noise and yields stable, high-signal attribution patterns. In our preliminary experiments, we observed that batch diversity, rather than predictor noise, was the primary determinant of attribution stability. By controlling sampling temperature and generating diverse batches, instability from models such as TemBERTure is effectively minimized over the course of optimization.
> * Regarding alternative attribution methods: approaches such as SHAP or causal-inference-style estimators were not adopted because they are computationally infeasible for our setting. SHAP’s complexity grows at least linearly with the number of masked feature combinations per sample (O(M×N)), and in the worst case exponentially with sequence length (O(2^N)). Given that our method requires attribution computation over hundreds to thousands of sequences across multiple optimization episodes, SHAP-based attribution would be computationally prohibitive.
> * In contrast, Integrated Gradients requires only a fixed number of gradient evaluations (O(k)), independent of input dimensionality, and satisfies key theoretical desiderata (completeness, sensitivity). Empirically, the noise introduced by IG is effectively averaged out through the sampling strategy described above, providing a practical and reliable signal at a fraction of the computational cost.
>
>
>
> **Q2**. The study mainly compares results with ProteinMPNN and HyperMPNN, which can be a bit simple and make it hard to situate the model’s performance relative to broader state‑of‑the‑art design frameworks. Please include more baselines such as RFdiffusion, MapDiff, ADFLIP.
>
> **Answer:**
> * We appreciate the request for broader baselines. Our choice of baselines was intentional and guided by the specific goal of this study: engineering hyper-thermostable sequences for a fixed backbone.
> * HyperMPNN is the only existing model explicitly trained for thermostability engineering:  our knowledge, no other generative model; including RFdiffusion, MapDiff, or ADFLIP has been trained to design high-temperature homologs. HyperMPNN is therefore the most relevant and strongest task-aligned baseline. Although we refer to it as a baseline, this does not diminish its strength; it is the SOTA model for thermostability optimization, and thus the appropriate target for direct comparison.
> * RFdiffusion is not an appropriate comparator for this task: RFdiffusion is a backbone generative model, designed for structure generation and conditional backbone scaffolding—not sequence optimization for a fixed backbone. It does not operate in the same problem setting as inverse folding models or our method, making direct comparison inappropriate.
> * MapDiff and ADFLIP have limited relevance due to low sequence diversity and lack of residue-level control: These models typically produce high-recovery, low-diversity sequences, as reflected in Table 1. Moreover, neither supports residue fixation, which is essential for realistic protein engineering tasks requiring preservation of catalytic, substrate-binding, or structural residues. In contrast, our framework allows explicit residue-level constraints.
> * PiFold was included specifically because it provides controllable sequence diversity: As shown, its temperature-based sampling allows a meaningful comparison of how diversity and foldability trade off across models.
>
> Additional benchmarks have now been included: For completeness and despite the above limitations we have added comparisons against additional state-of-the-art models in the revised manuscript. These results are now reflected in the updated tables and charts.
>
>
> In summary, while broad comparisons are valuable, the most meaningful benchmark for the thermostability-engineering task is HyperMPNN, and our method is evaluated directly against it. Broader baselines have been provided as requested.

---

> > ### Author Response · Authors · 2025-12-02
> >
> > **Q3**. The current metrics may not be enough for comprehensive evaluations. The author may expand the experimental validation dataset or integrate existing high‑throughput stability databases like FireProtDB; use broader biophysical measurements (e.g., unfolding rate, ΔH, or kinetic assays) to support multi‑dimensional validation; and introduce uncertainty quantification metrics (e.g., variance of attribution scores) to assess the reliability of explainability signals.
> >
> > **Answer:**
> > * Our evaluation already includes multiple orthogonal in silico biophysical proxies beyond predictor scores or attribution variance. Specifically, we perform (i) extensive conformational sampling via BioEmu, (ii) MD simulations, and (iii) MM/GBSA–based ΔG and ΔΔG calculations in OpenMM. As shown in the results, XPro-Design consistently produces variants with substantially lower ΔΔG than other methods. These energetic improvements correlate with known determinants of thermostability, including denser hydrogen-bond networks and shifts in core/surface amino-acid distributions—patterns also documented in empirical literature.
> > * On explainability reliability: the pairwise signed attribution agreement visualizations already quantify variance and stability of the IG-derived signals across sampled sequences. These provide direct insight into attribution consistency and highlight convergent optimization directions.
> > * Regarding integration with high-throughput stability databases (e.g., FireProtDB): these resources predominantly contain mesophilic data and are not aligned with the hyper-thermostability engineering task studied here. Using such datasets as ground truth for our design objective would introduce distributional mismatch and lead to misleading conclusions. Nevertheless, we agree that expanding experimental validation is important and plan to incorporate additional datasets and wet-lab measurements as the next phase of this project.
> > * In summary, while wet-lab biophysical assays are outside our current capabilities, the manuscript already includes a multi-dimensional, mechanistically grounded computational validation pipeline. The revised version clarifies these points to better reflect the breadth of our evaluation.
> >
> > Meanwhile, we are also working on securing the appropriate permissions to make the code base and data open source.

---

### Official Review · Reviewer_Mbn2 · 2025-10-31

**Soundness:** 2
**Presentation:** 2
**Contribution:** 2
**Rating:** 2
**Confidence:** 4

**Summary:**

The authors propose XPro-Design, a protein optimization framework that leverages pre-trained inverse folding methods and predictors to propose mutants with enhanced functionality. Additionally, Integrated Gradient is used to create the bias to temper the distributions used to choose positions and amino acids to mutate. Results for two proteins used as case studies show enhanced performance in different surrogate metrics.

**Strengths:**

1. Using the feedback from IG to bias inverse folding methods is interesting.
2. The authors validate the designs using various methods, ranging from traditional energy, predictors, structure predictors, and molecular dynamics.
3. Results show that for the case studies analyzed (one in the main text, one in Appendix), most metrics are enhanced for the candidates generated by the proposed method.

**Weaknesses:**

1. The methodology section is confusing, as it is hard to grasp how the framework in Fig. 1 can be implemented.
2. Only two case studies are investigated, and the setting might be unfair to baseline methods.
3. The writing has contradicting sentences. Additionally, the use of the term Explainable AI is arguable.
4. Code is not available.

**Questions:**

My initial recommendation is rejection due to the following reasons: (i) additional clarification is needed regarding the methodology, (ii) the writing has contradicting sentences throughout the manuscript, and (iii) the evaluation is limited and needs clarification regarding fairness.

Comments:

1. Figure 1 shows a loop that is repeated N times, but there are no details about this crucial part to understand the methodology in the main text.
2. In Table 1, there is a sequence recovery metric, but how many average mutations from the wild-type are for the candidates generated?
3. Given the framework in Figure 1, it seems that the PSSM matrix is learned in an active learning setting (with in silico predictors substituting wet lab experiments), but the baselines are just inverse folding methods, which raises questions about the fairness of the evaluation.
4. In Eq. 1 is P calculated by masking the residues individually and checking the probabilities given by ProteinMPNN and HyperMPNN?
5. (lines 181-183) There are no mathematical equations or references that support this statement.
6. How is Eq. 4 used by the proposed method?
7. Which predictor in Section 2.2 is used? Which are differentiable and which are non-differentiable?
8. In Table 2 is Boltz-2 or AF used? Compared to ProteinMPNN and HyperMPNN, the proposed method leads to the worst confidence metrics. Why do you think the proposed method is worse in these refoldability metrics?
9. How is the ddG in Fig. 4 calculated?
10. The conclusion has parts that contradict the characteristics of the proposed method. For example, “in a gradient free setting” and “while guiding optimization without task-specific predictors”. Additionally, the sentence in lines 483-485 does not reflect the current state in protein engineering and protein design research.

Minor Comments (that did not impact the score):

1. Many captions are missing a dot at the end. Also in line 199.
2. Using the term Explainable AI in this context, where IG is used to bias a distribution, needs proper support. This application seems more related to the definition of Interpretable ML in Murdoch et al “Definitions, methods, and applications in interpretable machine learning”.

---

> ### Author Response · Authors · 2025-12-02
>
> **Q1**. Figure 1 shows a loop that is repeated N times, but there are no details about this crucial part to understand the methodology in the main text.
>
> **Answer:**
> * The number of iterations N in Figure 1 is a user-defined hyperparameter. It is similar to the number of episodes in reinforcement learning, where N is determined by task-specific convergence rather than a universal preset. In our experiments, we ran the loop until rewards plateaued. More generally, users can set N based on their design goals or preferred stopping criteria.
>
> **Q2**. In Table 1, there is a sequence recovery metric, but how many average mutations from the wild-type are for the candidates generated?
>
> **Answer:**
> * Sequence recovery directly reports the fraction of positions that match the wild-type sequence. Consequently, a recovery value of 48% implies that approximately 52% of positions differ from the wild type.
>
> **Q3**. Given the framework in Figure 1, it seems that the PSSM matrix is learned in an active learning setting, but the baselines are just inverse folding methods, which raises questions about the fairness of the evaluation.
>
> **Answer:**
> * The evaluation is fair and intentionally conservative in favor of the baselines. In addition to P-MPNN, we benchmark against HyperMPNN, a SOTA inverse folding model fine-tuned specifically to generate thermostable homologs. This baseline therefore benefits from supervised training on curated thermostability data, whereas our framework does not leverage any supervised fine-tuning.
> * Despite this advantage, our method rapidly surpasses HyperMPNN, demonstrating that the proposed optimization loop is highly effective even without task-specific training data. This directly addresses the reviewer’s concern: the comparison is not only fair, but deliberately includes a strong, task-aligned baseline.
> * For completeness, we have expanded the revised manuscript to include comparisons with additional SOTA models.
>
> **Q4**. In Eq. 1 is P calculated by masking the residues individually and checking the probabilities given by ProteinMPNN and HyperMPNN?
>
> **Answer:**
> * No. P is not computed by masking residues individually. Instead, we estimate P empirically from multiple sampled sequences.
> * Concretely, for a given backbone, we run ProteinMPNN/HyperMPNN in standard sampling mode with all designable positions treated as mutable. In our experiments, we generate 1,000 sequences per backbone. We the compute the empirical amino-acid frequencies at each position, which yields a 20×L PSSM-like matrix P (protein length L).
> * When some residues are fixed, those positions are held constant during sampling, and the corresponding columns in P are effectively one-hot (probability of that position=1) while the remaining positions reflect the learned distribution of the model.

---

> > ### Author Response · Authors · 2025-12-02
> >
> > **Q5**. (lines 181-183) There are no mathematical equations or references that support this statement
> >
> > **Answer:**
> > * The statement “A key advantage of this approach is that model weights remain unchanged. This avoids catastrophic forgetting and prevents convergence to narrow local optima, while also preserving the structural fidelity of the inverse folding model; something that can degrade under fine-tuning on limited datasets.” is supported by extensive prior work showing that fine-tuning or task-specific training of inverse folding models can compromise foldability fidelity, particularly when optimizing for a single objective.
> > * In protein design, this failure mode is well documented:
> > * Stability–Function Trade-off:  As described in Engineering Strategies to Overcome the Stability–Function Trade-off, improving a specific functional or stability objective can push sequences past the threshold-robustness limit, after which folding stability collapses due to negative epistasis. This reflects the classical reward-hacking phenomenon: a model optimizing for thermostability alone may inadvertently generate variants with poor foldability unless an explicit foldability constraint is included.
> > * Evidence from recent protein-generation literature:
> > 	* ProteinZero (https://arxiv.org/html/2506.07459v2 ) highlights that optimizing a single reward signal without regularization leads to mode collapse and degradation of foldability. Its ablation studies show that removing the ESM-Fold–based foldability component significantly harms structural fidelity, explicitly demonstrating that fine-tuning without foldability supervision drives the model away from valid structural basins.
> > 	* RiboPO (https://arxiv.org/html/2510.21161v1) similarly incorporates folding-accuracy constraints into its reward function for RNA design, underscoring that structure-preservation is not automatically maintained during task-specific optimization.
> > 	* Additional evidence https://arxiv.org/html/2506.03028v1) further documents that unsupervised or reward-focused fine-tuning can distort the model’s generative manifold unless structural constraints are present.
> > * General ML phenomenon:  in ML it is well known that fine-tuning a pretrained model on a narrow task or small dataset can cause catastrophic forgetting or capability drift unless explicitly regularized. Inverse folding models exhibit the same behavior when their weights are updated to maximize a single reward signal.
> > * Our statement reflects this well-established phenomenon: once inverse-folding models begin updating their internal weights, foldability fidelity becomes vulnerable to drift, necessitating expensive structural reward terms (e.g., full folding models, regularizers, diversity constraints) to keep the optimization stable.
> > * In contrast, XPro-Design does not fine-tune model weights at all. The optimization occurs at the level of a low-rank bias matrix informed by averaged attribution scores. Because we never alter the inverse-folding model’s parameters, we avoid the foldability-drift failure mode documented above and consequently do not require expensive folding-model rewards or regularization.
> >
> > **Q6**. How is Eq. 4 used by the proposed method?
> >
> > **Answer:**
> > * Eq 4 is used to compute pair wise signed attribution scores which help us understand the epistatic interactions that contribute towards impact on the goal metric. This is analogous to epistatic heatmaps. Although this is not directly being used as part of the training reward, this is crucial to monitor trends over multiple iterations.
> > * In summary, a pairwise signed IG matrix is a functional epistasis map.
> > * It shows how the model believes amino acid positions interact to affect the prediction,
> > * revealing residue–residue synergy or antagonism.
> >
> > **Q7**. Which predictor in Section 2.2 is used? Which are differentiable and which are non-differentiable?
> >
> > **Answer:**
> > * We clarify that the primary predictor used within the optimization loop is TemBERTure (all three of its released variants).
> > * Sections 2.2 and 3.1 details this: these models provide differentiable predictions for Tm, which allows gradient-based attribution explanation in our framework.
> > * In addition, DeepSTABp is used outside the optimization loop for cross-model validation and for deriving consensus estimates of thermostability. This model is also differentiable, but they are not used to drive the optimization procedure itself. Only to evaluate consistency across independent predictors.

---

> > > ### Author Response · Authors · 2025-12-02
> > >
> > > **Q8**. In Table 2 is Boltz-2 or AF used? Compared to ProteinMPNN and HyperMPNN, the proposed method leads to the worst confidence metrics. Why do you think the proposed method is worse in these refoldability metrics?
> > >
> > > **Answer:**
> > > * Boltz-2 is used in Table 2 (the typo has been corrected). While our method yields slightly lower foldability confidence metrics than ProteinMPNN and HyperMPNN, these differences are marginal and remain well above standard acceptability thresholds for structure prediction.
> > > * A likely reason for this small drop is that our sequences diverge substantially from typical mesophilic sequences. AF-like models are predominantly trained on mesophilic distributions, and it is well documented that confidence can decrease when evaluating sequences that fall outside this distribution. Importantly, despite this distribution shift, the generated sequences still achieve high-confidence predictions, indicating that foldability is preserved.
> > >
> > > **Q9**. How is the ddG in Fig. 4 calculated?
> > >
> > > **Answer:**
> > > * The ddG computation follows the procedure described in Section 2.4. Briefly, for each designed variant, we sample 50 equilibrium conformations using BioEmu and then run molecular dynamics simulations followed by MM/GBSA calculations in OpenMM. The resulting MM/GBSA energies (reported as dG) are used as relative stability proxies. ddG values are obtained by taking pairwise differences of these dG estimates across variants. As noted in the manuscript, these values reflect relative stability trends rather than absolute folding free energies.
> > >
> > > **Q10**. The conclusion has parts that contradict the characteristics of the proposed method. For example, “in a gradient free setting” and “while guiding optimization without task-specific predictors”. Additionally, the sentence in lines 483-485 does not reflect the current state in protein engineering and protein design research.
> > >
> > > **Answer:**
> > > * The phrasing in the original conclusion was inadvertently ambiguous, and now corrected.
> > > * To clarify: “gradient-free” refers specifically to the optimization procedure. We do not update model weights and there is no backpropagation through the predictor or the IF model. Instead, we optimize by updating a biasing matrix using scalar attribution scores. Hence “gradient-free” in the sense of not using gradient-based parameter updates.
> > > * Regarding the phrase “without task-specific predictors,” this was a typo and has been corrected to “with task-specific predictors.” The revised conclusion now accurately reflects the role of TemBERTure as the task-specific stability predictor guiding the optimization loop.

---

### Meta-Review · Area_Chair_83Cq · 2026-01-06

**Summary:**

The submission presents XPro-Design, an AI-driven framework for protein engineering that integrates explainable components into a generative sequence design model targeting thermostability improvement. The authors claim improvements in property metrics such as melting temperature and folding free energy for designed variants relative to wild-type proteins.

**Reviewer Concerns:**

There is broad agreement that the paper’s main contribution is application-driven rather than methodological, with the proposed framework largely building on existing protein design and generative modeling approaches. Reviewers also note weaknesses in the experimental evaluation, including insufficient ablation studies and limited comparison to strong ML baselines. Also, the paper’s positioning remains unclear, as it does not convincingly demonstrate how the proposed ideas advance core machine learning research beyond a specific protein engineering use case. Although the authors provided some clarification, these concerns were not fully addressed, and I therefore lean toward rejection.

**Reviewer Scores:**

I do not expect any reviewer scores to change.

---

### Decision · Program_Chairs · 2026-01-26

Reject